

# Electroencephalogram-derived pain index for evaluating pain during labor

Liang Sun, Hong Zhang, Qiaoyu Han and Yi Feng

Department of Anesthesiology, Peking University People's Hospital, Beijing, China

## ABSTRACT

**Background**. The discriminative ability of a point-of-care electroencephalogram (EEG)-derived pain index (Pi) for objectively assessing pain has been validated in chronic pain patients. The current study aimed to determine its feasibility in assessing labor pain in an obstetric setting.

**Methods**. Parturients were enrolled from the delivery room at the department of obstetrics in a tertiary hospital between February and June of 2018. Pi values and relevant numerical rating scale (NRS) scores were collected at different stages of labor in the presence or absence of epidural analgesia. The correlation between Pi values and NRS scores was analyzed using the Pearson correlation analysis. The receiver operating characteristic (ROC) curve was plotted to estimate the discriminative capability of Pi to detect labor pain in parturients.

**Results**. Eighty paturients were eligible for inclusion. The Pearson correlation analysis exhibited a positive correlation between Pi values and NRS scores in parturients ($r = 0.768$, $P < 0.001$). The ROC analysis revealed a cut-off Pi value of 18.37 to discriminate between mild and moderate-to-severe labor pain in parturients. Further analysis indicated that Pi values had the best diagnostic accuracy reflected by the highest area under the curve (AUC) of 0.857, with a sensitivity and specificity of 0.767 and 0.833, respectively, and a Youden index of 0.6. Subgroup analyses further substantiated the correlations between Pi values and NRS scores, especially in parturients with higher pain intensity.

**Conclusion**. This study indicates that Pi values derived from EEGs significantly correlate with the NRS scores, and can serve as a way to quantitatively and objectively evaluate labor pain in parturients.

## INTRODUCTION

The pain experienced during labor has been described as one of the most painful events in a woman's life, and can be excruciating (*Chan et al., 2019*). The literature characterizes a vast array of factors which affect the perception of labor pain, including psychological mechanisms, duration of labor, fetus weight, and gene polymorphisms (*George, Allen & Habib, 2013*; *Terkawi et al., 2014*). Labor pain can also adversely impact the childbirth experience. In a severe pain scenario, labor pain can engender psychological stress, and in

Corresponding author
Yi Feng, yifeng65@outlook.com

some cases, negatively interfere with the normal process of labor, thereby increasing the rate of cesarean section (*Labor & Maguire, 2008*).

Timely and precise pain assessment plays a pivotal role in well-orchestrated pain management. Finding optimal ways to alleviate pain and thus improve both the physical and psychological well-being of people with pain is crucial to pain management (*Erden et al., 2017*).

To date, there are no widely accepted methods of objectively assessing pain, so a battery of subjective assessments such as the visual analog scale (VAS), the verbal rating scale (VRS), and the numerical rating scale (NRS) are generally used (*Kaibori et al., 2015*). These subjective scales, however, are affected by subjective influences, evaluation lag, and many other factors. Accordingly, a wide range of objective physiological indicators including heart rate variability, blood pressure variability, nociception level (NoL) index, analgesia nociception index (ANI), skin electrical conductivity, and other possible indicators for pain monitoring have been developed (*Stöckle et al., 2018*; *Edry et al., 2016*; *An et al., 2017*). However, these pain-monitoring indicators are also affected by a broad spectrum of factors and thus generate disappointing results when trying to accurately assess pain. Therefore, a more reliable objective, quantitative indicator is urgently need.

Several noninvasive imaging devices merit consideration (*e.g.*, positron emission tomography, functional magnetic resonance imaging); however, as argued in the literature, there is still disagreement about the degree to which current measures of brain activity exactly relate to pain. Given the high temporal resolution of electroencephalograms (EEGs) and their ability to capture real-time changes within the brain, they are regarded as the most promising neuroimaging devices for an objective pain diagnosis (*Benoit et al., 2017*; *Okolo & Omurtag, 2018*; *Reches et al., 2016*; *Levitt & Saab, 2019*).

The Pain index (Pi; Beijing Easymonitor Technology Co., Ltd, Beijing, China), a novel point-of-care indicator, was developed from the whole frequency band-EEG wavelet algorithm. A previous study confirmed that Pi significantly correlated with NRS scores in chronic pain patients, and thus provided valuable insights into the possibility of objective pain assessment in clinical settings (*An et al., 2017*). Labor pain differs significantly from chronic pain. According to its definition, labor pain consists of two discomfort dimensions: a sense of physical pain intensity and a psychological stress state. Moreover, parturients undergo different trajectories and mechanisms of pain during the first and second stages of labor (*Kafshdooz et al., 2019*). Specifically, the pain during the first stage of labor is largely visceral in origin and cannot be localized well, whereas during the transitional and second stages, the somatic pain becomes more intense and well located in the lower part of the abdomen (*Yuksel et al., 2017*). The reliability and validity of Pi for assessing labor pain in the obstetric setting require further clarity and because of the nature of labor pain, parturients are a natural model for a study of Pi that includes both visceral and somatic pain. Building on these considerations, we extended our line of investigation to identify the correlation between Pi values and NRS scores in parturients, as well as explore the discriminative ability of Pi to detect both visceral and somatic pain.

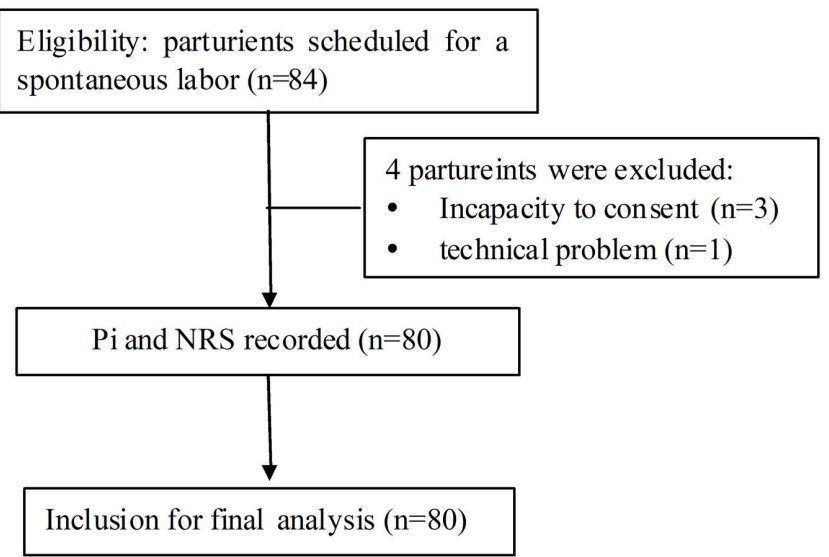

**Figure 1** **The flow diagram of the study protocol.** Pi, pain index; NRS, numerical rating scale.

## METHODS

This study was approved by the Institution Review Board (Ethics Committee of Peking University People's Hospital, Peking University, No. 2016PHB031-01) and registered on the Chinese Clinical Trials Registry (ChiCTR1800014769). Informed written consent was obtained from all participants. The flow chart of study participation was shown in Fig. 1.

### Study procedure

Parturients who were admitted to the delivery room at the Department of Obstetrics, Peking University People's Hospital between February and June of 2018 were consecutively enrolled, and epidural analgesia was implemented according to the requirements of the parturients.

Inclusion criteria were: (1) women who were 21–39 years old; (2) had a gestational age of 37–42 weeks; (3) were prepared to deliver vaginally with a single, cephalic, term pregnancy; (4) parturients who entered the first or second stage of labor; (5) and were admitted to the delivery room during daytime working hours (between 8 am to 5 pm).

Exclusion criteria included: (1) a history of psychiatric disease (those who were diagnosed before or during pregnancy by a psychiatrist); (2) or those incapable of giving consent.

Demographic and clinical data were collected including: maternal age, height, weight, gestational age, parity, stages of labor, and epidural analgesia information. Since pain complaints are usually reported by the parturient during the first and second stage of labor, and since the third stage lasts for a very short time (usually less than 30 min) and is usually accompanied with mild pain, we did not collect the relevant data during the third stage. In addition to blood pressure and oxygen saturation as measured by pulse oximetry ($SpO_2$), real-time Pi values were collected from a multifunction monitor (Beijing Easymonitor Technology Co., Ltd, Beijing, China) with acquisition electrodes. Parturients were asked to

**Figure 2 Placement of EEG electrodes and display of Pi.** (A) Three electrodes were on each subject's forehead: one was 2 cm above the midpoint between the eyebrows and two were above the bilateral eyebrows; (B) the two reference electrodes were mounted on the bilateral mastoid processes (right one is shown here); (C) real-time display of the WLI and the Pi on the integrated EEG monitor, with the values updating every 2 s (sampling rate: 1,600 Hz). Pi, Pain index; WLI, wavelet index.

quantify their pain intensity using a 0-to-10-point NRS (with "0" representing no pain and "10" the worst pain; an NRS score ≤ 3 corresponds to mild pain, 4–6 to moderate pain, and 7–10 to severe pain (*Boonstra et al., 2016*)) scale at different time points (initiation, apex and interval of three consecutive uterine contractions) during either the first or second stage of labor aided by a tocometer. If the parturient refused to describe their pain due to severe pain, the NRS score was permitted by a subsequent recall when the pain subsided. The Pi values (0–100) at the above-mentioned time points were also recorded.

## EEG collection and Pi calculation

EEG signals were simultaneously monitored and collected by 5 electrodes, according to the manufacturer instructions and a previous study (*Wang et al., 2020*). There were 3 electrodes on each subject's forehead: one located 2 cm above the midpoint, and two placed above the bilateral eyebrows. Meanwhile, two reference electrodes were placed on the bilateral mastoid processes (Figs. 2A, 2B). The raw EEG signals were collected and further processed for the calculation of Pi values.

The EEG analysis software package (Beijing Easymonitor Technology Co., Ltd., Beijing, China) was applied, based on the whole frequency band EEG wavelet algorithm, which is currently one of the most suitable tools for analyzing EEGs (*Zhang et al., 2010*; *Constant & Sabourdin, 2012*). The algorithm formulas and calculation methods of raw EEG data processing were discussed previously (*An et al., 2017*), and the details can be found in Supplemental Information 2. Pi values were displayed on the monitor screen in a real-time manner (Fig. 2C), with values updating every 2 s (sampling rate: 1600 Hz). Moreover, taking into account that EEGs are susceptible to the impact of the electromyography (EMG), EMG components were filtered during Pi value calculation.

## Statistical analysis

A sample size of $n = 70$ parturients was calculated according to a minimal detectable correlation between Pi values and NRS scores of $r = 0.4$, with an α error of 5% and a power of 80%. In total, we included 84 participants to allow for an approximate 20% loss due to protocol violation, parturient withdrawal, or technique-related problems.

**Table 1  The clinical characteristics of included parturients ($n = 80$).**

| Variables | Values |
|---|---|
| Age (years) | $32.1 \pm 4.0$ |
| Height (cm) | $162.3 \pm 4.8$ |
| Weight (kg) | $73.0 \pm 12.1$ |
| Gestational age (day) | $274.7 \pm 15.8$ |
| Primipara n (%) | 70 (87.5%) |
| Epidural analgesia n (%) | 21 (26.3%) |
| First stage of labor n (%) | 60 (75%) |
| Second stage of labor n (%) | 20 (25%) |

Notes.
  Data are expressed as mean $\pm$ standard deviations (SDs) or frequencies (percentages).

Data are presented as the mean $\pm$ standard deviations (SDs) for normally distributed continuous variables or median [interquartile range] if distributions were skewed, while categorical variables are expressed as frequencies or percentages. Normal distribution of the data was tested using the Kolmogorov–Smirnov test. The degree of dependency between variables was estimated using the Pearson correlation test. In addition, the receiver operating characteristic (ROC) curve was applied to test Pi's ability to discriminate between subjects who will and those who would develop moderate or severe labor pain (NRS > 3) and those who would not. The area under the curve (AUC) estimates were also determined in order to indicate the probability of accurately discriminating between the different pain groups. The Youden index was calculated and defined as the value of "sensitivity + specificity-1". In general, the optimal cut-off value was calculated according to the ROC curve analysis, maximizing the Youden index (*Youden, 1950*). The SPSS 19.0 (SPSS Inc., Chicago, IL, USA) package was used, and a $P < 0.05$ was considered statistically significant.

## RESULTS

### Clinical characteristics of parturients

A total of 80 parturients fulfilled the inclusion criteria and were enrolled into the final analysis; four patients were excluded because of either an incapacity to consent ($n = 3$) or a technical problem ($n = 1$). Detailed demographic and clinical characteristics of all parturients were shown in Table 1. Finally, the NRS scores and Pi values at the same time points were averaged for further analysis.

### Correlation of Pi value with NRS score in parturients

The Pearson correlation test showed a positive correlation between Pi value and NRS score in the enrolled parturients ($r = 0.768$, $P < 0.001$, Fig. 3), suggesting that Pi could be used for pain monitoring in parturients.

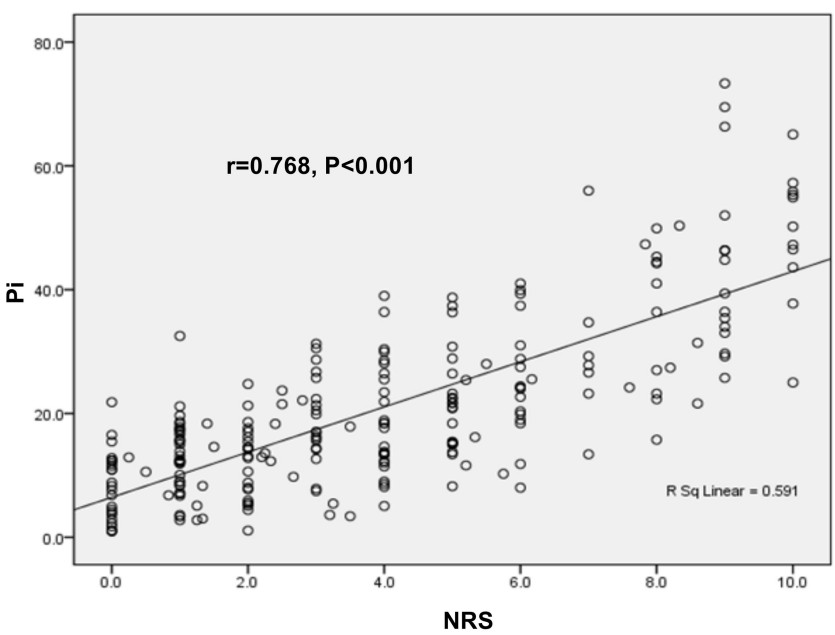

**Figure 3  Correlations of Pi value with NRS score in parturients.** The expression of Pi value was positively correlated with NRS score ($r = 0.768$, $P < 0.001$).

## Subgroup analyses for correlation of Pi values with NRS score controlling for different confounding factors

In order to further substantiate the correlation of Pi values with NRS scores, subgroup analyses were performed taking the underlying confounding factors into account.

### a. Different stages of labor

During the different stages of labor, the correlation between Pi values and NRS score were as follows: a. the first stage of labor ($r = 0.741$, $P < 0.001$); b. the second stage of labor ($r = 0.774$, $P < 0.001$) (Figs. 4A and 4B).

### b. The presence or absence of analgesia

The Pearson correlation test that considered the possible impact of epidural analgesia also exhibited a positive correlation between Pi value and NRS score: c. the presence of analgesia ($r = 0.760$, $P < 0.001$); d. the absence of analgesia ($r = 0.750$, $P < 0.001$), as shown in Figs. 4C and 4D.

### c. Different time points of uterine contractions

As for uterine contractions, the correlation between Pi value and NRS score varied between the different time points as follows: e. the initiation of uterine contraction ($r = 0.582$, $P < 0.001$); f. the apex of uterine contraction ($r = 0.751$, $P < 0.001$); g. the interval of uterine contraction ($r = 0.487$, $P < 0.001$), as shown in Figs. 4E, 4F and 4G. These results suggested that the correlation between Pi values and NRS score was stronger at the apex of uterine contractions compared to the initiation and interval of uterine contraction.

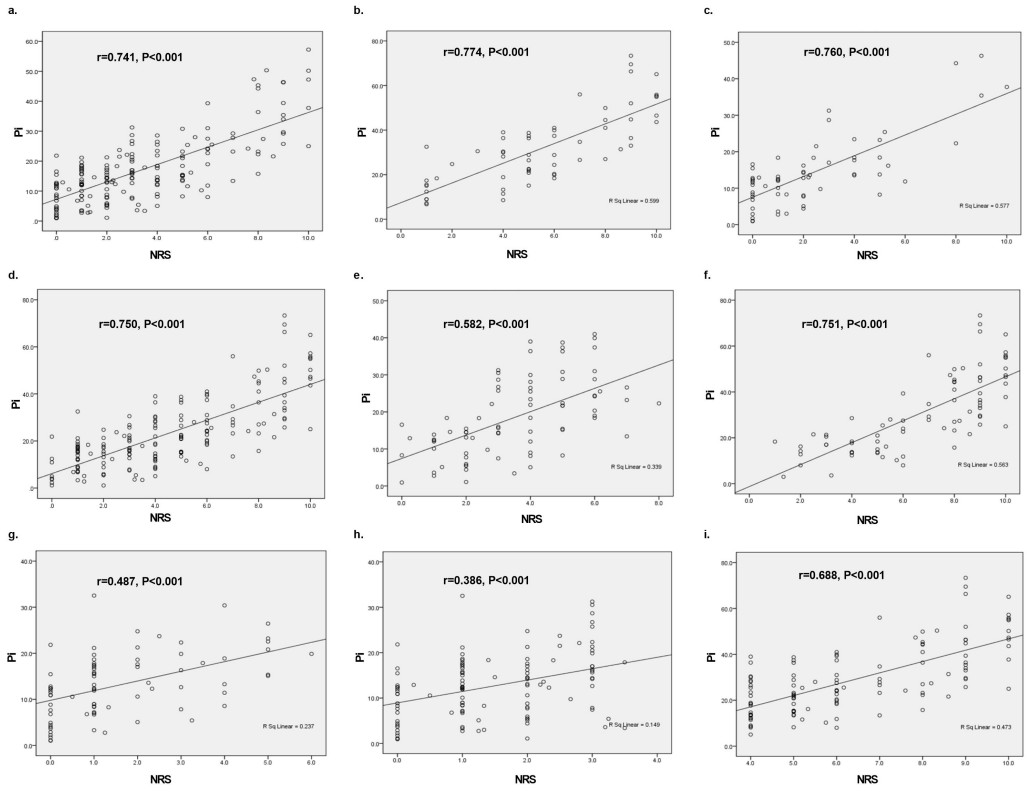

**Figure 4  Subgroup analysis on correlations of Pi value with NRS score during labor based on different confounding factors.** (A) The first stage of labor ($r = 0.741$, $P < 0.001$); (B) the second stage of labor ($r = 0.774$, $P < 0.001$); (C) the presence of analgesia ($r = 0.760$, $P < 0.001$); (D) the absence of analgesia ($r = 0.750$, $P < 0.001$); (E) the initiation of uterine contraction ($r = 0.582$, $P < 0.001$); (F) the apex of uterine contraction ($r = 0.751$, $P < 0.001$); (G) the interval of uterine contraction ($r = 0.487$, $P < 0.001$); (H) NRS score $< 4$ ($r = 0.386$, $P < 0.001$); (I) NRS score $\geq 4$. ($r = 0.688$, $P < 0.001$).

### d. Pain intensity

As discussed previously, an NRS score $< 4$ corresponds to mild pain and an NRS score $\geq 4$ indicates a moderate or severe pain. With respect to pain intensity, the Pearson correlation test showed a stronger correlation between Pi value and NRS score in parturients with moderate or severe pain compared to those with mild pain: h. NRS score $< 4$ ($r = 0.386$, $P < 0.001$); i. NRS score $\geq 4$. ($r = 0.688$, $P < 0.001$), as shown in Figs. 4H and 4I.

### Ability of Pi to discriminate between mild and moderate-to-severe labor pain

According to the ROC analysis, we evaluated the capability of Pi to distinguish between mild and moderate-to-severe labor pain in parturients. The ROC curve constructed using Pi revealed that the AUC was 0.857 (Fig. 5A), and the optimal cut-off value of Pi was 18.37, with a sensitivity, specificity and Youden index of 0.767, 0.833 and 0.6, respectively. Moreover, when the cut-off value was set to 15.0, the sensitivity was the highest (0.825). There was, however, a lower specificity and Youden index of 0.633 and 0.458, respectively (Table 2).

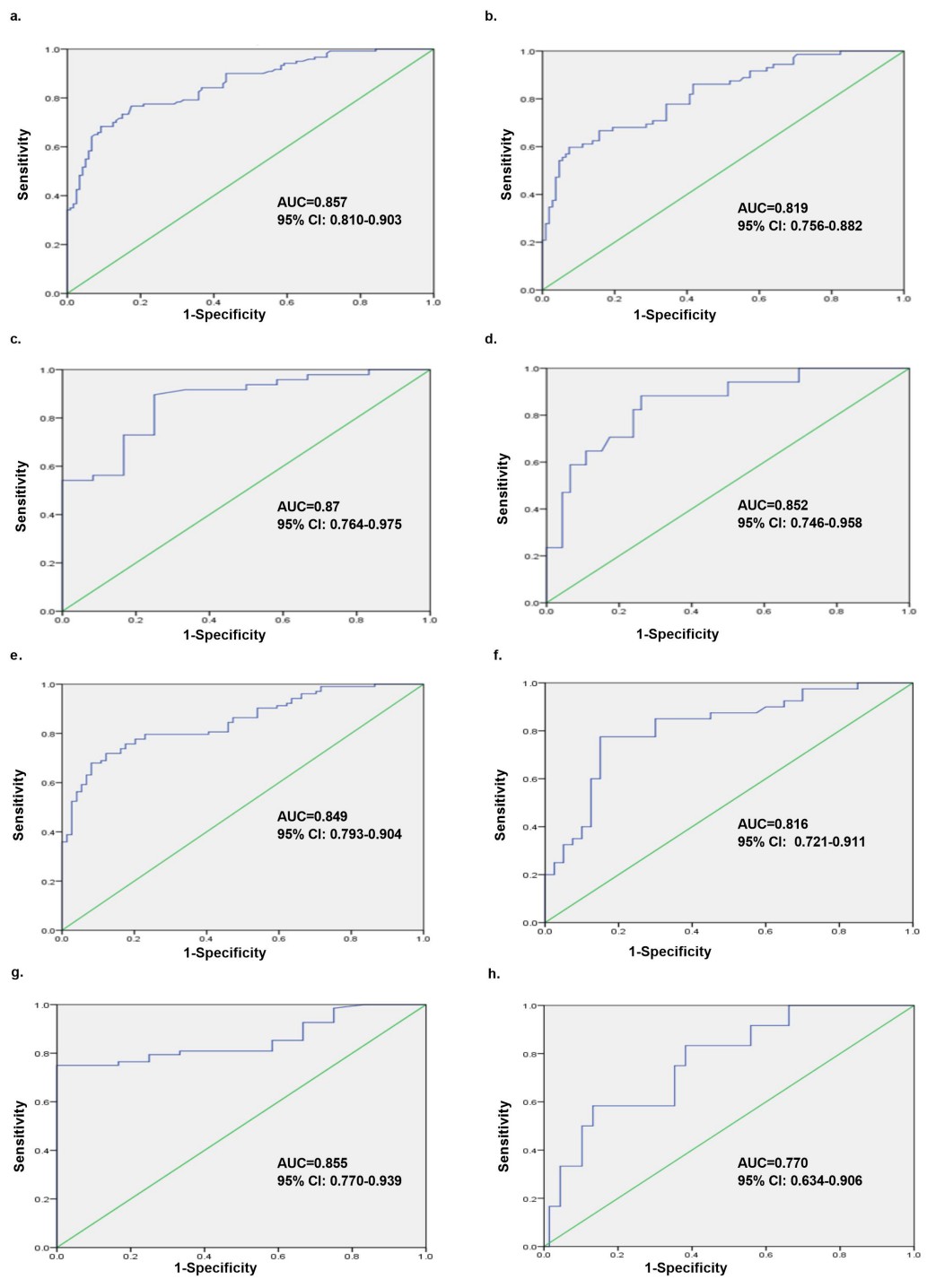

**Figure 5** **Overall analysis and subgroup analyses of Pi's ability to discriminate between mild and moderate-to-severe labor pain in different scenarios.** The ROC analyses were performed for the Pi in labor pain subjects, and Pi could distinguish between 

**Figure 5 (...continued)**
mild and moderate-to-severe labor pain subjects with different AUCs. (A) Overall analysis containing all the parturients (AUC = 0.857, 95% CI [0.810–0.903]); (B) the first stage of labor (AUC = 0.819, 95% CI [0.756–0.882]); (C) the second stage of labor (AUC = 0.87, 95% CI [0.764–0.975]); (D) the presence of analgesia (AUC = 0.852, 95% CI [0.746–0.958]); (E) the absence of analgesia (AUC = 0.849, 95% CI [0.793–0.904]); (F) the initiation of uterine contraction (AUC = 0.816, 95% CI [0.721–0.911]); (G) the apex of uterine contraction (AUC = 0.855, 95% CI [0.770–0.939]); (H) the interval of uterine contraction (AUC = 0.770, 95% CI [0.634–0.906]). ROC, receiver operating characteristic; AUC, area under the curve.

**Table 2** **The capability of different Pi cutoff values to distinguish labor pain in parturients.**

| Cutoff | Sensitivity | Specificity | Youden index |
|---|---|---|---|
| 15.00 | 0.825 | 0.633 | 0.458 |
| 16.08 | 0.783 | 0.700 | 0.483 |
| 16.99 | 0.775 | 0.742 | 0.517 |
| 18.37 | 0.767 | 0.833 | 0.600 |
| 19.31 | 0.717 | 0.850 | 0.567 |
| 20.11 | 0.708 | 0.867 | 0.575 |
| 21.07 | 0.683 | 0.875 | 0.558 |
| 22.01 | 0.650 | 0.917 | 0.567 |

**Notes.**
The Youden index was calculated and defined as the value of "sensitivity + specificity-1". The optimal cut-off value was calculated, maximizing the Youden index.

## Subgroup analyses of ability of Pi to discriminate between mild and moderate-to-severe labor pain in different scenarios

We also performed subgroup analyses to strengthen the diagnostic ability of Pi to discriminate between mild and moderate-to-severe labor pain in parturients.

### a. Different stages of labor

The ROC analysis also demonstrated that Pi could distinguish between mild and moderate-to-severe labor pain in parturients at different stages of labor: b. the first stage of labor (AUC = 0.819, 95% CI [0.756–0.882]); c. the second stage of labor (AUC = 0.87, 95% CI [0.764–0.975]), as shown in Figs. 5B and 5C.

### b. The presence or absence of analgesia

Taking into account the impact of analgesia, the ROC analysis showed that Pi could distinguish between mild and moderate-to-severe labor pain in parturients with or without epidural analgesia: d. the presence of analgesia (AUC = 0.852, 95% CI [0.746–0.958]); e. the absence of analgesia (AUC = 0.849, 95% CI [0.793–0.904]), indicating that there was no influence of analgesia on the diagnostic performance of Pi (Figs. 5D and 5E).

### c. Different time points of uterine contractions

The ROC analysis considered the time points of uterine contraction and revealed that Pi could distinguish between mild and moderate-to-severe labor pain in parturients at different time points during uterine contractions. The highest diagnostic value was at the apex of contraction: f. the initiation of uterine contraction (AUC = 0.816, 95% CI

[0.721–0.911]); g. the apex of uterine contraction (AUC = 0.855, 95% CI [0.770–0.939]); h. the interval of uterine contraction (AUC = 0.770, 95% CI [0.634–0.906]) as shown in Figs. 5F, 5G and 5H.

## DISCUSSION

Our results from the current study support the notion that Pi values, extracted from EEGs, exhibited a positive correlation with NRS scores in parturients. Following a previous study conducted in a chronic pain setting (*An et al., 2017*), the diagnostic accuracy of Pi was powerful enough to apply the cut-off value clinically (the sensitivity, specificity and Youden index were 0.767, 0.833 and 0.6, respectively).

Although complex modulation is implicated in the pain conduction pathways, in theory, EEGs can collect information on neuronal activation, evaluate the components of pain activity, and thus provide a feasible method for processing complex pain information (*Wagemakers et al., 2019*; *Levitt & Saab, 2019*). However, the bispectral index (BIS), an EEG-based variable widely used in the general anesthesia setting, can only provide invaluable inference for the level of consciousness of the patients and so fails to serve as an accurate indicator of nociceptive stimulation. Currently, there is no advanced, reliable, objective and quantitative way to measure pain in clinical practice (*An et al., 2017*). In this study, we collected EEG signals from the prefrontal lobe using two channels (left and right) and further extracted the pain components, exhibiting the reliability and validity of Pi for diagnosing labor pain.

Evidence suggests that pain can cause significant and specific changes to multi-brain regions and multi-frequency EEG signals (*Roberts et al., 2008*; *Mouraux & Iannetti, 2018*). *Li et al. (2016)* conducted continuous EEG recordings on 19 healthy subjects and found a strong correlation between the placebo effect on reported pain perception and alpha amplitude, which suggested that alpha oscillations in frontal-central regions serve as a cortical oscillatory basis of the placebo effect on tonic muscle pain. Likewise, significant overactivation of the pain network was detected in multiple brain regions (*e.g.,* parietal lobule, anterior cingulate, thalamus, anterior and posterior insula, dorsolateral prefrontal cortex (DLPFC) and S1) in a large population of chronic neuropathic pain patients using quantitative EEG, which was in line with conventional functional neuroimaging findings and extended to cover the mid and posterior cingulate, supporting the notion that the enhanced temporal resolution of electrophysiological methods may facilitate more precise identification and evaluation of the pain network (*Prichep et al., 2018*). Similarly, correlations of pain perception and functional alterations in the brain (mainly identified using EEG) have also been implicated in other pain settings, including visceral pain and fibromyalgia (*Mayer et al., 2015*). A recent study investigating a novel pain recognition indicator, the pain threshold index (PTI), which is based on the same principle as Pi, revealed a better predictive accuracy for postoperative pain than the surgical pleth index (SPI) and could differentiate moderate-to-severe pain from mild pain in patients who had recovered from general anesthesia. Therefore, together with our findings, the current evidence lends increasing support for the potential role of EEG as an objective pain indicator, which includes indicators extracted from EEG like Pi.

In addition, if available, epidural analgesia remains a highly requested modality for pain relief during labor (*Traynor et al., 2016*), and performing pain evaluations during epidural analgesia is a big concern. Therefore, we performed subgroup analyses taking confounding factors into account (*e.g.*, stages of labor, analgesia, uterine contractions and pain intensity), and our results further substantiated a robust correlation between Pi values and NRS scores, positing that Pi is suitable for evaluating the labor pain of parturients. Specifically, our study showed a stronger correlation and higher diagnostic ability in parturients with a higher pain intensity (those with NRS $\geq$ 4 or at the apex of uterine contraction) than those with mild pain. Markedly, we calculated a cut-off value of 18.37 for predicting labor pain under the optimal Youden index. However, considering the unique characteristics of labor pain, in combination with the principle of early diagnosis and treatment, we strongly recommended the cut-off value of Pi be set at 15.0, with the highest sensitivity (0.825), even with a lower specificity.

Our study has several limitations. First, our study population was relatively small, and more reliable results could likely be obtained with more participants. Second, we omitted some potential confounding factors from our study (*e.g.*, fetus weight, the third stage of labor, *etc.*), possibly causing risk bias, which merits future investigation. Third, we employed a tocometer to judge the time points of initiation, termination, and apex of each uterine contraction. We believe that the tocometer is currently the most accurate method for judging these time points, but some may disagree. For future research, a well-designed, large-scale, multi-center study is still needed to further validate the feasibility of Pi in an obstetric setting.

## CONCLUSION

In summary, the current study confirmed that Pi, based on the EEG wavelet algorithm, as a noninvasive, objective, point-of-care monitoring indicator can reflect the existence and the severity of pain in parturients. Pi might serve as a potential alternative in the future to help clinicians quantitatively and objectively assess labor pain.

### Funding

This work was supported by the National Key Research and Development Program of China (2018YFC2001905). The funders had no role in study design, data collection and analysis, decision to publish, or preparation of the manuscript.

### Grant Disclosures

The following grant information was disclosed by the authors:
The National Key Research and Development Program of China: 2018YFC2001905.

### Competing Interests

The authors declare there are no competing interests.

## Author Contributions

- Liang Sun performed the experiments, analyzed the data, prepared figures and/or tables, authored or reviewed drafts of the paper, and approved the final draft.
- Hong Zhang and Yi Feng conceived and designed the experiments, authored or reviewed drafts of the paper, and approved the final draft.
- Qiaoyu Han performed the experiments, analyzed the data, authored or reviewed drafts of the paper, and approved the final draft.

## Human Ethics

The following information was supplied relating to ethical approvals (i.e., approving body and any reference numbers):

The study was approved by the Institution Review Board (Ethics Committee of Peking University People's Hospital, Peking University) (2016PHB031-01).

## Data Availability

The raw data are available in the Supplemental Files.

## Supplemental Information

Supplemental information for this article can be found online at http://dx.doi.org/10.7717/peerj.12714#supplemental-information.

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
