# Peer review of "Electroencephalogram-derived pain index for evaluating pain during labor"

_PeerJ, doi:10.7717/peerj.12714_

## Round 0.1 · original submission · Major Revisions

Dear Authors,

There are major revisions to be done with your manuscript as per comments from the peer reviewers.

·

Basic reporting

The article is written in sufficiently clear English that can be understand by vast population.

Experimental design

The experiment seek to find the correlation between Pain index (Pi) and NRS score rating for pain due to labor.

Validity of the findings

There is indeed some correlation of the Pi and NRS measured in this experiment. However, in my humble opinion the correlation is not strong and may be due to the noises recorded in the EEG, especially when the pain in question is a result of muscles contraction (which caused the recorded EMG). According to the article by An et al 2017b, the Pi is a derivative from the weighted wavelets coefficients, these coefficients may increase in value when there is a strong EMG recorded due to muscles contraction. This effect is also seen during some of the apex of labor contraction under epidural analgesia when no painful response is recorded in the NRS. The authors need to make sure that appropriate filtering of the EEG signals is applied to remove or minimize the influence of EMG. Nonetheless, I think there is some usefulness of this study as it covers a reasonable numbers of samples of about 1100 samples from 80 subjects. The author should explain the above limitation in the discussion section.

Additional comments

no comment.

Reviewer 2 ·

Basic reporting

The authors address a clear goal of this paper in which they intended to come up with more reliable and quantitative measures of pain during labor. In this study, the authors claimed to develop a pain index that is derived from the frequency band EEG wavelets. In general, the flow is logical and structured accordingly. However, some improvements can be made in all of the figures and tables provided. The unit of measurement for Pi and NRS should be mentioned elsewhere in the main text and also in both exes of the graphs. No title of the table and this makes it hard for the reader to make sense of the content of the table. Presentation-wise of the table title should be improved.

Experimental design

Lack of details in EEG recording and data analysis. The authors claim that they used EEG as a device to come up with EEG frequency spectrum and develop the pain index. Lacking the details of EEG pre-processing and frequency analysis may question the validity and reliability of the findings. That information should be mentioned clearly in the method so that others can replicate it as well.

Apart from that, the authors should not miss out on the details of their EEG recording device such as sampling rate, etc. area of the electrode position, following the standard 10-20 EEG system position.

Validity of the findings

The authors need to justify of only including two stages of labor in this study. If there are other stages, should state in the literature.

In terms of sample size, the authors stated that their findings are less reliable due to the small sample size. Please state a required sample size and show the calculation for enough samples that can make their findings becoming more reliable.

In lines 269-270, the authors mentioned that they might exclude some confounding factors. This is a very ambiguous statement. Please be more specific and state what are the factors they are referring to as their limitation.

---

## Round 0.2 · accepted · Accept

Dear Authors, Feel free to add the references suggested by one of the peer reviewers if they are relevant to your manuscript.
I would like to suggest some references of similar study in which the author may refer to:

SAI, C.Y., MOKHTAR, N., YIP, H.W. et al. Objective identification of pain due to uterine contraction during the first stage of labour using continuous EEG signals and SVM. Sādhanā 44, 87 (2019).

Kumar S, Kumar A, Trikha A and Anand S. Electroencephalogram based quantitative estimation of pain for balanced anaesthesia. Measurement 59: 296–301 (2015).

·

Basic reporting

Manuscript is clear and easy to understand.

Experimental design

Experiment design is simple and easy to understand.

Validity of the findings

Results are statistically sound although not of significant scientific advancement.

Additional comments

I would like to suggest some references of similar study in which the author may refer to:

SAI, C.Y., MOKHTAR, N., YIP, H.W. et al. Objective identification of pain due to uterine contraction during the first stage of labour using continuous EEG signals and SVM. Sādhanā 44, 87 (2019).

Kumar S, Kumar A, Trikha A and Anand S. Electroencephalogram based quantitative estimation of pain for balanced anaesthesia. Measurement 59: 296–301 (2015).